# Family-Friendly Policies: Extrapolating A Pathway towards Better Work Attitudes and Work Behaviors in Hong Kong

**DOI:** 10.3390/ijerph191912575

**Published:** 2022-10-01

**Authors:** Lina Vyas, Francis Cheung, Hang-Yue Ngo, Kee-Lee Chou

**Affiliations:** 1Department of Asian and Policy Studies, The Education University of Hong Kong, Tai Po, Hong Kong; 2Department of Applied Psychology, Lingnan University, Tuen Mun, Hong Kong; 3Department of Sociology, The Chinese University of Hong Kong, Shatin, Hong Kong

**Keywords:** family-friendly policies, work attitude and behavior, supervisory family support, mediator

## Abstract

The need for family-friendly policies to balance work and life demands is growing. Many studies have addressed how family-friendly policies relate to a variety of employees’ work attitudes and behavioral outcomes, but not how they (positively or negatively) affect them, especially the affective components of family-friendly policies that provide “felt” support to an employee. To fill this gap, this study adopts a moderated mediating mechanism to analyze how affective components of family-friendly policies impact employees’ attitudes and behaviors through signaling and social exchange theory. We examined how this impact is mediated by factors such as work–life conflict, perceived organizational support, and control over working hours, as well as whether having a supportive supervisor moderates the mediated effect through further limiting the degree of work–life conflict or strengthening control over working hours. Data were collected through a survey with 401 employee–supervisor dyads from organizations in Hong Kong. We found that family-friendly policies do not necessarily affect work attitude and behavior, but they work through the sequential mediators of having more control over working hours and perceived organizational support. The role of supportive supervisors is also significant, in that they are likely to be key in molding the organizational environment for the gradual provision and uptake of family-friendly policies. The results of this study contribute to the development of signaling and social exchange theory and have theoretical implications for supervisors regarding them utilizing their position to improve employee work attitudes and behavioral outcomes.

## 1. Introduction

Long working hours are common among Hong Kong employees, with about 33.7% working more than 48 h per week [1]. While all European Union member states and the UK require employers to ensure that their employees do not exceed 48 h per week including overtime and have sufficient daily and weekly breaks [2], currently there is no statutory law in place in Hong Kong that protects employee rights in terms of balancing work and family demands [3]. Studies have shown that work–life conflict (WLC)—which is defined as an inter-role conflict whereby the participation in one role (e.g., work) causes difficulties in the participation in another (e.g., family)—is particularly severe in Hong Kong [4].

Reducing WLC may lead to various occupational outcomes, such as job satisfaction [5], and familial outcomes, such as life satisfaction [6]. To avoid WLC, the research of it has emphasized the importance of family-friendly policies (FFPs; e.g., gender-differentiated effect of FFPs on work–life balance [7]; a review of the policy gap by Skinner and Chapman [8]). Interestingly, there is a lack of research intending to delineate the mechanism by which FFPs prevent WLCs. In this study, the goal is to extrapolate the pathways by which FFPs lead to changes in work attitudes and behaviors by examining WLC, perceived organizational support, and control over working hours as mediating factors of it, and a supportive supervisor as a moderating factor of it.

### 1.1. Family-Friendly Policy

Taking responsibility for enhancing a work–life balance depends heavily on the culture and the employees of an organization. Although policies can be implemented at a higher level to help balance work and life demands (e.g., labor laws protecting employee rights and instituting fixed working hours), most of the everyday WLC is handled at the organizational level, such as emergency family leave which is granted to employees by their immediate supervisors. According to Lazar, Osoian, and Ratiu, work–life balance practices in organizations are deliberate changes in the programs or organizational culture that reduce WLC and enable employees to be more effective in their various roles [9]. Organizations use a variety of ways to improve the work–life balance, including by providing flexible working hours, telecommunications (working from home), compressing the work week, part-time work schedules, and job sharing, amongst others. Such approaches to reducing WLC can be captured in one term, “family-friendly policy.”

An FFP is a potential alleviator of WLC, and it is related to a myriad of different employee attitudes and behavioral outcomes in many contexts and different sectors of the population. It has been shown that an FFP is directly or indirectly (as a mediator) associated with increased job satisfaction in Asia [10,11,12] and globally [13,14], as well as in different career categories, such as for federal employees [15], and those in different life situations, particularly for working mothers [12,13]. In addition to job satisfaction, an FFP is also related to an increased commitment to organizations [10,16,17,18] and intention to stay in the workplace [19,20,21]. Improved measures of job engagement [22] and task performance [11,23] have been observed in organizations providing FFPs to employees, as has a reduced degree of withdrawal from work [24].

### 1.2. Hong Kong as a Case Study

With it having some of the most demanding work cultures and longest hours among advanced economies, the average working week in 2018 was 42 h in Hong Kong, which was longer than it was in South Korea, Japan, the United States, the United Kingdom, and Germany [25]. It is difficult for employees in Hong Kong, particularly those with family responsibilities, to strike a harmonious balance between work and life. Local surveys have shown that the majority of the city’s employees are stressed at work and experience severe WLC [3]. One recent study suggested that the estimated annual economic losses that were caused by work-related stress could be up to HK$7.09 billion, considering the costs of absenteeism, workplace conflicts, and medical expenses [26]. While the health and well-being of the working population in Hong Kong are matters of concern, the city is still at an early stage of FFP development [3].

To date, most of the research on FFPs has been carried out in the US and in European countries; whereas much less attention is given to Asia, despite the region being home to a majority of the world’s workforce. In addition, few studies on employees’ work attitudes and behaviors have been conducted in Asia, in general, and in Hong Kong in particular. The reality that many employees are facing WLC is exacerbated by the work culture and conditions that are found across Asia, leading to a situation of potential concern given that many are denied family-friendly employment policies. This study examines the issues of employees’ work attitudes and behavioral outcomes, and how FFPs can assist them in resolving conflicts between work and family in Hong Kong, which is a unique case for working conditions in Asia.

### 1.3. Research Question: How Do Family-Friendly Policies Affect Work–Life Conflict?

Although a perusal of the literature shows that the availability and utilization of FFPs is linked to work attitudes and behavior [27,28], the mechanism through which these effects occur has not been adequately explored. To address this limitation, the first objective of this study is to uncover the potential pathways through which affective components of an FFP impacts employees’ work attitudes and behavioral outcomes. From the perspective of signaling and social exchange theories, WLC and control over working hours can mediate the impact of FFPs on work attitudes and outcomes, which are potentially moderated by the effect of having supportive supervisors. A comprehensive understanding of this moderated mediating mechanism is critical to advancing research and planning on FFPs. In addition, this study seeks to shed light on the way that employees expect organizations to help them reduce WLC and offer control over working hours. The second objective of this study is to make the case for the types of FFP that are context-appropriate based on the analysis of the theorized pathways, with practical application in other countries.

## 2. Theory Building and Hypothesis Development

### 2.1. Signaling Theory: The Mediating Role of Work–Life Conflict and Perceived Organizational Support

One dominant mode of thinking is that an FFP works through signaling theory [28,29]. According to signaling theory, employees may form impressions of their employers’ unobservable motives by interpreting their observable actions. For example, if the organization allocates financial resources to employee training [30] or provides FFPs [28], then employees may perceive their employer as being committed to maintaining their long-term employment. It is believed that the availability of FFPs improves work attitude and behavior through signaling the perception of a family-supportive organization to employees [27,28]. The availability of FFPs is interpreted by employees as one of many *signals* that their organization cares about them [27,28] and their families, and they consequently perceive the organization to be family-supportive [16].

However, in reviewing the literature on family-friendly practices and organization performance, Beauregard and Henry found that such practices do not necessarily reduce employees’ WLC [31]. That is to say, the availability of an FFP alone does not signal to employees that their organization is supportive. For example, without a conducive environment in which to utilize an FFP, or without the active promotion of the FFP by supervisors or HR management, the signaling pathway is weak as employees do not feel its direct benefits. As such, besides the “cognitive” component of knowing that there is the availability of an FFP, there is also an “affective” or “felt” component of an FFP—employees sense that their organization is supportive, which results in affecting their attitude and behavior. Other studies, instead of focusing directly on employee attitudinal or behavioral outcomes, have explored these other “felt” components. We categorize these “felt” components into two broad categories, depending on the various interactions of employees with their work–life conditions.

One interaction is employees’ work–life interface. Studies have shown that the provision of FFPs reduces WLC by allowing employees to better manage the workplace and home demands [32,33,34,35]. Some have studied WLC as a mediating factor between FFP availability/utilization and work outcomes [27]. In addition, specific FFP provision such as flexible working hours gives employees more control [36,37], thereby often alleviating home and work pressure, particularly for women in Asia, who may traditionally be responsible for domestic work. As such, control over working hours is often identified as a mediating link between FFP and improved work outcomes, such as lower levels of stress, higher levels of commitment to employers, and overall reduced organizational costs [38,39].

Since WLC and control over working hours can be considered to be mediators between the awareness of FFP use and its outcomes, we propose that they are the two main signals to employees that the organization is supportive. When employees feel that they have control over their working hours or a reduced WLC as a result of FFPs, they will perceive the organization to be supportive, which is a sequential mediating effect. In turn, this affects their work attitude and the behavioral outcomes (the top left portion of Figure 1 highlights these relationships).

Accounting for these components of signaling theory, two hypotheses will be tested to better understand how FFPs are associated with employee attitudes and behaviors. We hypothesize that:

**Hypothesis** **1.**
*Work–life conflict and the perception of a supportive organization are the sequential mediators of the FFP’s availability and utilization, thereby affecting work attitudes and behaviors.*


**Hypothesis** **2.**
*Control over working hours and the perception of a supportive organization are the sequential mediators of FFP’s availability and utilization, thereby affecting work attitudes and behaviors.*


### 2.2. Social Exchange Theory: The Moderating Role of Supportive Supervisors

Another felt component and interaction is that which is between the employer and the employee. Social exchange theory [40] indicates that the relationships between employers and employees are based on both economic and non-economic exchanges and are important to employee outcomes [41]. In fact, various kinds of social interactions are closely intertwined with employee outcomes, such as organizational commitment, turnover intention, and withdrawal behavior [42]. However, there are no assessments in the literature as to whether the provision of FFPs influences the kind of interactions that employees have with their employers, for example, in social exchanges. An extension of the discussion of the employee–employer interface and their social exchange lies in the role of the supervisor. A supportive supervisor who facilitates employees having a positive work–life balance [43] has been shown to be often associated with work-related outcomes [16,44] including their work attitude [45] and job performance [46]. For instance, supervisors who express concern and offer encouragement to employees who are experiencing high WLC tend to care about employees’ family obligations and offer flexibility to help them strike a work–life balance. Not surprisingly, supervisory family support is strongly associated with job satisfaction and their intention to stay in the role [47], as well as with work behavior [48]. Moreover, De Simone et al. found that a work–life balance is positively associated with job satisfaction [49]. Supervisors also have an impact on employee outcomes by creating an environment in which FFPs can be established. Employees who take advantage of an FFP may not be fully satisfied or relieved from WLC if they are afraid of experiencing a backlash and negative career consequences by profiting from of such a policy [50]. For example, some supervisors may believe that policy users are less committed to the organization, prioritize family over career, and increase the workload of the supervisors [16]. However, if the supervisor is family-supportive, the benefits of FFP use are fully actualized. Thus, we propose that family support from supervisors may guarantee a positive effect of policy use and minimize any anticipated negative effect (the bottom left portion of Figure 1 highlights this relationship).

An FFP that is used in a supportive-supervisor environment is associated with lower WLC and more control over working hours—indicating to employees that the organization is supportive, and thereby, impacting employee behavior and outcomes.

**Hypothesis** **3.**
*Supportive supervisors enhance the impacts of FFPs (if available) on employee attitudes and behaviors by limiting WLC or strengthening their control over working hours.*


## 3. Methods

### 3.1. Participants and Procedure

A multistage random sampling method was adopted to select an entirely new sample of respondents satisfying the following criteria: (1) aged 25 or above; (2) residing and working in Hong Kong; (3) employed on a full-time basis during the research period; and (4) working under an immediate supervisor. At first, 1000 companies were randomly selected from the Hong Kong Yellow Pages. A letter was sent to each of the companies setting out the purpose of the study, and a follow-up phone call was made to solicit their participation. Those companies that agreed to participate were asked to nominate an employee–supervisor pair for the study. If there was more than one potential respondent pair in a company, then it was randomly selected for survey following the “next birthday” rule. Four well-trained and experienced research assistants conducted the 30-min face-to-face survey with the participants on-site using a structured questionnaire. To ensure the daily quality of the data, the research assistants were closely monitored by a fieldwork supervisor. The investigators randomly checked 10% of all of the completed questionnaires.

### 3.2. Measures

*Availability and utilization of family-friendly policy*: The survey covered 18 family-friendly benefits, such as flexible working hours and paternal leave [33,51], and participants were asked to indicate which benefits were provided by their employers and whether they had utilized them. A global question capturing the availability and utilization of these was used as the main predictor in this study: “Does your company provide organizational policies to enable employees to balance their work and non-work life? If such benefits were available, would you utilize them?”

*Work attitudes:* Four dimensions of job attitude were measured: job satisfaction, affective commitment, intention to stay, and work engagement. Job satisfaction was assessed by a 3-item scale which was adapted from previous studies [52], and affective commitment was assessed by an 8-item scale [53]. Participants were asked to answer both scales on a 7-point Likert scale ranging from 1 (strongly disagree) to 7 (strongly agree). The three items of job satisfaction were: “All in all I am satisfied with my job,”, “In general, I don’t like my job,”, and “In general, I like working here,”, while a sample item on the affective commitment scale was “I feel a strong sense of belonging to my organization.” We measured the intention to stay with a 3-item scale which had been used in previous studies [54,55], and participants answered on a 5-point Likert scale ranging from 1 (strongly disagree) to 5 (strongly agree). The three items were “Barring unforeseen circumstances, I would remain in this organization indefinitely,”, “If I were completely free to choose, I would prefer to continue working in this organization,”, and “I expect to continue working as long as possible in this organization.” Lastly, a 9-item scale which was adopted from a recent study [56] was used to measure job engagement, in which participants were asked to report how they felt about their job on a 7-point Likert scale ranging from 1 (never) to 7 (always). A sample job engagement item was: “At work, I feel that I am bursting with energy.”

*Work behaviors*: Work behaviors include task performance, organizational citizenship behavior (OCB), citizenship behavior towards the supervisor, and job withdrawal. Task performance was assessed using a 7-point scale from 1 (strongly disagree) to 7 (strongly agree) which was developed by Williams and Anderson [57]. OCB comprises the behaviors that aim to support other individuals in the organization (OCBI) and the organization as a whole (OCBO). Citizenship behavior towards the supervisor was measured with a 5-item scale, which was rated on a 5-point scale ranging from 1 (never) to 5 (very often) [58]. Job withdrawal is an aspect of job behavior, and this was measured by an 8-item psychological withdrawal behavior scale [59]. Participants were asked to report the frequency with which they had engaged in withdrawal behavior in the past three months on a 5-point scale ranging from 1 (never) to 5 (always). A sample item was: “Daydreamed or allowed your thoughts to wander on the job.”

*Work–life conflict.* WLC was measured with a Chinese version of a 10-item work-to-family and family-to-work conflict scale [60] which was developed by Netemeyer, Boles, and McMurrian [61]. Participants were asked to answer questions that were graded on a 7-point Likert scale ranging from 1 (strongly disagree) to 7 (strongly agree).

*Perceived control over working hours*: Perceived control over working hours was measured with a 7-item scale [43]. Participants were asked to rate how frequently they encountered this in their work on a 5-point scale ranging from 1 (very little) to 5 (very much). One sample item was: “How much choice do I have over the amount and timing of work I must do at home in order to meet my employment demands?”

*Perceived family supportive organization:* The extent to which the organization was perceived to be family-supportive was measured with a 9-item scale which was adopted from a previous study [44]. Participants were asked to rate the extent to which they agreed or disagreed with the items on a 7-point Likert scale ranging from 1 (strongly disagree) to 7 (strongly agree). A sample item was: “My organization makes an active effort to help employees when there is a conflict between work and family life.”

*Supervisory family support:* Supervisory family support was assessed with a 4-item scale which was developed in a recent study [45]. Participants were asked to rate their agreement with items on a 5-point Likert scale ranging from 1 (strongly disagree) to 5 (strongly agree). A sample item was: “My immediate supervisor works effectively with employees to creatively solve conflicts between work and non-work or family issues.”

*Control variables*: We included three control variables, age, gender, and education level, as the working experiences and outcomes were likely to differ within these dimensions.

### 3.3. Analytical Strategy: PROCESS Macro 

The analysis used the PROCESS macro in SPSS which was developed by Hayes to estimate the multiple effect pathways in the complex mediation models [62]. This strategy was chosen over other modeling techniques such as structural equation modeling due to the direct measurement of variables from the questionnaire (i.e., the assumption that a lack of latent variables is present).

For each variable (exposure, mediator, outcome of work attitude, and behavior), the average score across all of the questions measuring the concept was taken after ordering the questions in the same qualitative direction. This has the effect of transforming the Likert-type scales into a quantitative scale. This method is not recommended by some methodological schools as a means to simplify the interpretation of a multiple mediator model because it breaks the ordinal nature of the Likert scale and decategorizes meaningful groups; given the many exposure–mediator–outcome pathways that are involved, however, and the small data set, transforming the Likert scale into a quantitative scale variable turns it into a spectrum for easier interpretation. The focus of the analysis highlights the general trend of association and not the quantum of change that occurs as a result of exposure.

### 3.4. PROCESS Macro, Mediation, and Moderated Mediation 

Figure 1 presents the serial mediation model. The pathway is structured so that exposure is associated with an indicator, which in turn associates with the perception of a supportive organization, which associates with the outcome. The PROCESS model produces estimates for all theoretically possible combinations of pathways from exposure to outcome in order to estimate the total effect that exposure is exerting on the outcome via the different pathways. Figure 1 (right) displays the statistical diagram of the tests that were conducted. In the second set of hypotheses, this serial mediation pathway changes when a supportive supervisor interacts with an FFP, thereby “moderating” how the availability and use of an FFP impacts the relationships. In other words, the relationship of FFPs to the first indicator is tested to see whether having a supportive supervisor affects the indicator. This will, in turn, affect the perception of the organization, which therefore affects the outcome. Since this model constitutes both the serial mediation pathway and moderation, it is also referred to as moderated serial mediation.

## 4. Results

A total of 401 dyads of employees and their immediate supervisors were successfully surveyed, yielding a response rate of 47.1%. The mean age of the employees was 38 (SD = 7.9), 66% were women and 91% were married, and they worked an average of 50 h per week (SD = 8.8). The cross-correlation matrix and Cronbach’s alpha scores are presented in Table 1. 

In the first set of hypotheses, a serial mediation analysis was performed to examine the mechanisms that could link FFP provision and use to work attitude and behavior. The salient results are presented both in statistical form (Table 2 and Table 3) and in graphical form (Figure 2).

First, the FFP indicator is associated with increasing control over working hours (0.42, *p* < 0.001), but it has no significant association with a reduction in WLC (−0.007, *p* = 0.95). However, these two variables were strongly associated with the perception of a supportive organization, with those who had less WLC (−0.1, *p* = 0.02) and more control over working hours (0.44, *p* < 0.001), thereby viewing their organizations as supportive.

Second, the indicators were associated with most of the work attitudes and behaviors that were examined in this study. An increase in WLC was associated with a decreased degree of affective commitment (−0.16, *p* < 0.001), a decreased intention to stay (−0.19, *p* < 0.001), and an increased degree of work withdrawal (0.07, *p =* 0.004). Control over working hours was associated with *all* types of work attitudes: job satisfaction (0.26, *p* = 0.002), affective commitment (0.33, *p* < 0.001), intention to stay (0.29, *p* < 0.001), and job engagement (0.18, *p* = 0.04). For all of the outcomes, apart from citizenship behavior, the perceived support of an organization was the most important variable which was associated with the various outcomes. In Figure 2, all of the boxes that are shaded purple and dark gray indicate positive associations of both the indicator and perceived support of an organization. The light-blue boxes indicate the association of the perceived support of an organization, only. It is important to note is that no outcomes are directly related to FFP provision and use, with only citizenship behavior being an associated outcome of this. This association is negative, as having and using FFPs is associated with lower citizenship behavior when one is considering the four different indicators (−0.16, *p* = 0.04 in the WLC model; −0.23, *p* = 0.005 in the control over working hours model). The lack of a direct association between FFPs and their outcomes other than citizenship behavior will be explored in the discussion section.

Third, the proposed serial mediation pathway—whereby an FFP associates with the indicators, thereby affecting the perceived supportive status of an organization, and thus affecting the different work attitudes and outcomes—differed based on the factors that were observed. Since the FFP has no insignificant effect on WLC, but it has a positively significant effect on control over working hours, Hypothesis 1 is not supported and Hypothesis 2 is supported. The *individual* effects are shown in Table 2 and Table 3 for their easier interpretation.

Hypothesis 3 concerns a moderated mediation model to see whether having a supportive supervisor changes any of the relationships between FFP use and both the first indicator (WLC, control over working hours) and the outcomes. The tests indicate whether having supportive supervisors magnified or reduced FFP provision and use in the pathway. Assuming that increased supervisor support would enhance FFP use, this would then be associated with the mediators (reduced WLC, more control over working hours), thus improving the perceived support of the organization, and associating it with the various outcomes. The results are presented in Table 4 and Figure 3. First and foremost, having a supportive supervisor is associated with reduced WLC (−0.80, *p <* 0.001) and greater control over working hours (0.19, *p* = 0.005). However, the interaction of an FFP and a supportive supervisor is insignificant, suggesting that supportive supervisors are not significantly related to FFP provision and use, and thus, they do not impact the overall model pathway, thereby allowing us to reject Hypothesis 3. This does not, however, detract from the important point that supportive supervision is closely associated with all of the outcomes.

## 5. Discussion and Implications

This research shows that the availability and utilization of FFPs influences the attitudes and behaviors of employees towards work. This study hopes to add to the literature by presenting empirical testing of this phenomenon in an East Asian population. We looked at whether FFP provision and utilization operate through two identified indicators (i.e., WLC and control over working hours) to allow employees to perceive the organization as supportive. This perceived support of an organization affects a myriad of work-related attitudinal and behavioral outcomes. The study also examines whether supervisory support interacts with the affective components of FFPs to the extent that stronger support from supervisors amplifies their impact. Since the conceptual models propose that the variables work in tandem to associate with the outcomes, the “weak link” in these associations requires us to reject two out of three proposed hypotheses. Still, there are important bilateral relationships between the variables that should be explored in greater detail.

Hypotheses 1 and 2 concern the link between the availability and use of FFPs and work attitudes and behaviors, through the mediation pathway of a reduced WLC and an increased control over working hours, thereby leading to the perceived supportiveness of an organization. We found that FFP availability and utilization was associated with an increasing control over working hours, as opposed to reducing WLC. This suggests that the scope of WLC exceeds that which the current limited design of the FFPs can handle. Supporting this point is Skinner and Chapman’s research by it showing how flexible work schedules may not be suitable for all employees [8]. They describe flexibility intervention as “beneficial when it involves an increase in workers’ control over work scheduling, such as self-rostering shift-work.”

A significant association is observed between both of the indicators (WLC and control over work hours) and a perceived supportive status of the organization. This relationship is strongest in the mediation pathway. The theoretical proposition is that signaling theory must work through a “felt” component of a reduced WLC and control over working hours due to FFP use. When the employees experience the “felt” components of an FFP, they feel supported by the organization, and perceive the organization to care about their overall well-being. As a consequence, they will develop a positive relationship with the organization, becoming emotionally attached and committed to the organization [63]. This results in an improved and positive job attitude, and it enriches the employees’ work performance and commitment.

In effect, the provision and use of FFPs signals the support of the organization to an employee by giving them more autonomy in their work–life balance, and the employee responds through a set of reciprocal obligations [64]. For example, an employee’s experience of having a lessened degree of WLC may lead to the perception that the organization cares about their work and family responsibilities, thereby leading the employee to understand that the organization will help them to manage work and family responsibilities [64], which ultimately benefits both the organization and employees [65]. This finding suggests that the perception of a supportive organization is the main mediating mechanism through which family-supportive benefits affect the organizational outcomes [16]. In our model, the perceived support of an organization is the most important indicator in the pathway, which is associated with nearly all of the outcomes. As described by the signaling theory, those that are perceiving the organization to be supportive are likely to show a better work attitude and behavior by simply enjoying the workplace more [16,28].

The organizations that we surveyed had some sort of FFP in place, or at least some flexibility in work arrangements, but there were mostly basic options such as five-day working weeks. It is possible that there is some underutilization of FFPs, which could be mistaken for under-provision. There are a few reasons for the occurrence of underutilization. The first is that family-supportive policies might be implemented without a needs assessment or a needs analysis having been conducted (a “misprovision”). According to Pitt-Catsouphes and Bankert, a needs-based assessment is crucial for an organization to determine which family-friendly initiatives are relevant to their employees, thereby potentially avoiding the implementation of initiatives that employees do not perceive as important [66]. For instance, Donnelly, Proctor-Thomson, and Plimmer reported that although the majority of workers (female public servants in their case) benefited from flexible work practices and reported that they had control over the timing of holidays and breaks, most of them lacked control over the amount of work or overtime hours that they completed, thereby resulting in negative outcomes in the work sphere, such as reduced satisfaction and commitment [67]. Therefore, the FFPs that are provided may be poorly tailored to some employees in the office.

The second reason is that employees might not be well informed of the work–life policies and benefits that are available in their workplace, thereby resulting in the underutilization of such benefits [68]. This is supported by the fact that the average number of FFPs that are available exceeds the average number of FFPs that had been previously used. This can be easily rectified through better human resource education on the availability of these policies in the organization. 

Third, there might not be sufficient informal workplace support, such as family-supportive supervision, or a family-supportive organizational culture [16,69]. Valcour and Batt suggested that if the employees don’t perceive their supervisor as supporting their family-work balance, they are less likely to use the family-friendly programs that are provided in the workplace [70], so as not to be perceived negatively by their supervisor. This is captured in our results, and it is perhaps a macroscopic symptom of the work culture of Hong Kong, and possibly of Asia more broadly. The findings indicate that FFP provision and its use is associated negatively with citizenship behavior through the variable assessing citizenship behavior towards supervisors from the supervisor perspective. It is therefore also possible that the uptake or utilization of FFPs might actually give a worse impression to the supervisor, because FFP uptake signals that an employee lacks responsibility, or does not take up challenges.

The finding that FFPs are not associated directly with any work attitude or behavioral outcome, apart from citizenship behavior towards the supervisor, suggests that FFP utilization is counter-effective in the Hong Kong context. These findings are supported by other studies [69,71], thereby indicating that many employees do not make use of work–life benefits because of fears and reasonable anticipation of this having negative career consequences. For instance, it is a common perception that female workers do not take maternity leave because they are convinced that such career breaks will harm them professionally [71], as such, the current public policy design on the length of maternity leave is still trapped in the 1980s-type thinking. In contrast to the prospect of employees enjoying the benefits of an FFP, there was a perception of a deterioration in their role, value, and career advancement prospects which are caused by the use of an FFP that would allow them to be away from the organization for a time. Related research has also noted a dilemma amongst employees using FFPs, particularly when they perceive their supervisors to be unsupportive of the policy. Likewise, Lobel and Kossek stated that work–life policies would not achieve the desired results unless the organizations achieved genuine change in organizational norms and values regulating the appropriate role of non-work considerations in the workplace [72]. In a context where an FFP may not be widely distributed or accepted, this highlights a deeper structural issue in the economic transitions in Asian culture, which cherishes diligence and self-reliance as opposed to a more liberal workplace that values shared success, teamwork, and smart working.

This situation is further exacerbated by the relationship between supervisors and employees, as explored in the last hypothesis. Having a supportive supervisor is an important component in reducing WLC and gaining control over working hours. One reason that is attributed to this finding is that the supervisors play an integral role in establishing and fulfilling their employees’ work-based psychological contract [73]—essentially, this is a specific version of social exchange theory. Direct supervisors are responsible for managing and evaluating their employees’ performance, and for conveying their decisions to upper management. Therefore, employees tend to regard the attitude and behavior of their supervisors as an indication of the organization’s level of support [74]. In other words, a direct supervisor is an employee’s key representative in an organization, interpreting and enacting explicit and implicit organizational exchanges, thus serving as an exchange partner. The nature of Chinese culture means that a high-quality exchange relationship between the employee and the supervisor is an important aspect of the workplace in Hong Kong [75]. If tacit signals suggest that FFP uptake is looked upon unfavorably in the organization, communicated directly or indirectly through their immediate supervisors, FFP uptake can be discouraged. Thus, an FFP environment—which is fostered by the supervisor—should strengthen the association between the FFP and its outcomes.

The assumption that the FFP indicator captured the availability and utilization of the FFP as one conditional term is the main limitation of this study. It takes groups of people who enjoy both the availability and utilization of FFPs and compares them with others who have neither. In future studies with larger samples, it is necessary to distinguish between the availability and utilization of FFPs to disentangle the differences between these two concepts, and it is important to explore the different conditions under which their utilization may occur. In addition, the way in which FFPs are measured may produce different results. Some studies examine only one policy or measure, whereas others explore a list of policies or measures [76]. This difference is important because there may be a dosage effect in the availability or utilization of FFPs [77]. Moreover, some researchers argue that FFPs are not homogeneous and can be classified into two primary categories: support measures providing resources for dependent care such as childcare centers, and flexibility measures providing alternative work locations or hours (e.g., flexitime or home office). Therefore, the wide range of FFPs that are available may affect the outcomes that are measured. Future studies could explore how different types of FFP affect the outcomes, which may also have implications for their suitability within companies.

## 6. Conclusions

Using a data set from Hong Kong, this paper examines how FFPs influence employee attitudes and behaviors through signaling and social exchange theory. Three hypotheses have been examined to better understand how FFPs are connected to employee attitudes and behaviors in Hong Kong.

Our study demonstrates that a reduced WLC and more control over working hours are closely linked to the perceived support of the organization, and subsequently, to work attitudes and behavioral outcomes in an East Asian setting. The role of FFPs is found to be marginal, but this may be due either to lack of availability and/or utilization of them within the general population. We show that supervisors play a meaningful part in establishing how employees perceive and experience their organization’s environment, and hence, they may be responsible for establishing a “high-quality working environment” that is apt for FFP utilization. Thus, supervisors must understand the power they have in shaping company dynamics and strive to develop high-quality exchange relationships with their employees, which will ultimately lead to organizational growth. Based on the above rationale, organizations in Hong Kong should continue to offer supervisors training on employees’ work-family balance and the appropriate administration of work-family benefits [16]. This would establish strong social exchange relationships between the employees and their organization, thereby signaling the organization’s support to employees, in order to cultivate long-term trust-based relationships with staff. This may include delivering FFPs as a tool for supervisors to strengthen their relationships with employees, whether they are legally provided or mandated by organizations. This will also help employees to cope with competing work–life demands. However, it is up to the supervisors to “set the tone” and decide whether FFP utilization is appropriate and should be made available to an employee, as without such considerations, their provision will be little better than having no provision at all.

The results of this study contribute to the development of signaling and social exchange theory and have theoretical implications for supervisors regarding that utilization of their position to improve employee work attitudes and behavioral outcomes. The results prove that signaling theory must work through a “felt” component, that is tangible to the employee, of a reduced control over working hours that is due to FFP use. Supervisors, playing an integral role in establishing and fulfilling their employees’ work-based psychological contracts, also proved the impact of social exchange theory. In practice, organizations or employers should strengthen the association between FFPs and employees’ outcomes.

## Figures and Tables

**Figure 1 ijerph-19-12575-f001:**
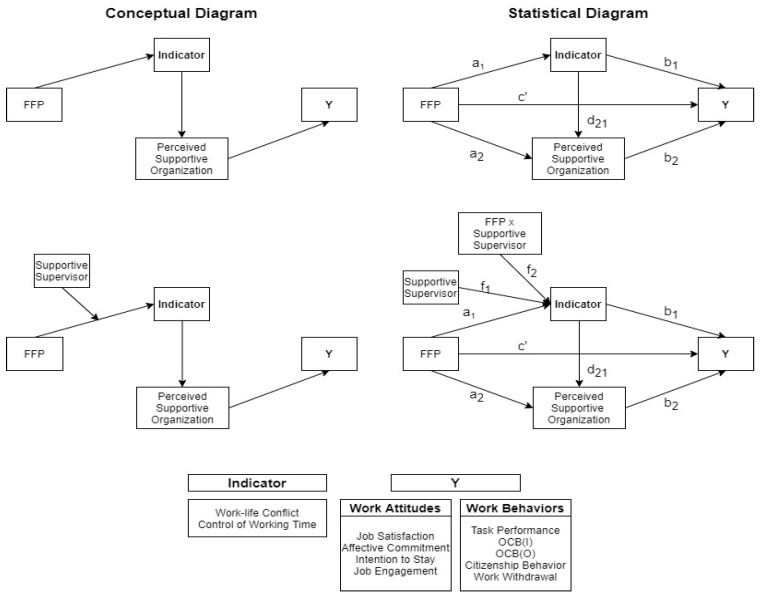
Conceptual and statistical diagrams for analysis of family-friendly policy and supervisor support on various outcomes through serial mediation pathways.

**Figure 2 ijerph-19-12575-f002:**
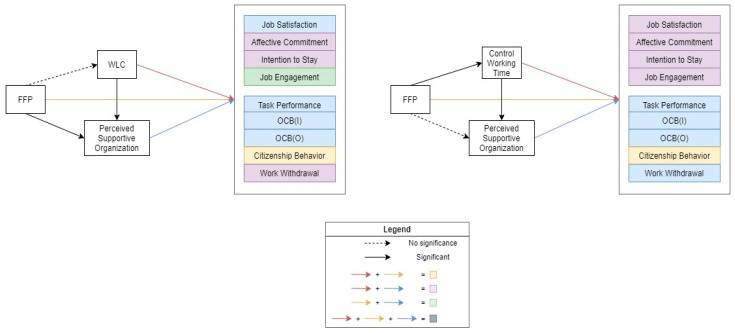
Association of family-friendly policies with work attitude and behavioral outcomes through the serial mediator model.

**Figure 3 ijerph-19-12575-f003:**
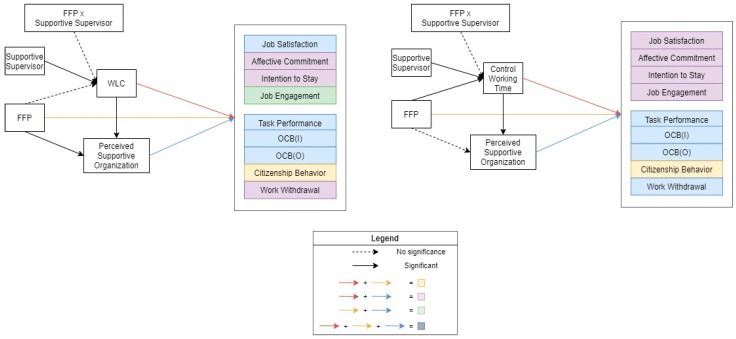
Association of family-friendly policies with work attitude and behavioral outcomes through the moderated serial mediator model.

**Table 1 ijerph-19-12575-t001:** Descriptive statistics and intercorrelations among study variables.

Variable	M	SD	1	2	3	4	5	6	7	8	9	10	11	12	13	14	15
Work–life conflict	23.87	10.58	(0.93)														
Perceived control over working hours	12.19	4.07	0.04	(0.86)													
Perceived family-supportive organization	26.15	9.15	−0.05	0.26 *	(0.94)												
Supervisory family support	9.35	2.43	−0.36 *	0.20 **	0.37 *	(0.86)											
Number of family-friendly policies available	4.13	3.21	0.10 *	0.35 **	0.28 *	0.03											
Number of family-friendly policies used before	2.55	2.20	0.05	0.36 **	0.26 *	0.09	0.83 **										
Job satisfaction	11.66	2.76	−0.25 **	0.17 **	0.16 *	0.24 *	0.05	0.07	(0.77)								
Affective commitment	28.46	7.83	−0.10	0.30 **	0.33 *	0.33 *	0.16 **	0.13 **	0.59 **	(0.92)							
Intention to stay	7.26	2.38	−0.26 **	0.26 **	0.27 *	0.38 *	0.14 **	0.13 *	0.58 **	0.77 **	(0.90)						
Job engagement	29.07	9.20	−0.11 *	0.22 **	0.36 *	0.26 *	0.18 **	0.23 **	0.45 **	0.58 **	0.44 **	(0.95)					
Task performance	18.46	3.20	−0.06	−0.05	0.11 *	0.16 *	−0.02	−0.05	0.34 **	0.29 **	0.24 **	0.23 **	(0.94)				
OCB(I)	19.96	5.24	−0.01	0.07	0.13 *	0.15 *	−0.05	−0.01	0.23 **	0.16 **	0.10 *	0.29 **	0.40 **	(0.90)			
OCB(O)	17.93	5.76	−0.04	0.09	0.21 *	0.12 *	0.03	0.03	0.26 **	0.29 **	0.19 **	0.34 **	0.30 **	0.58 **	(0.91)		
Citizenship behavior	12.98	3.77	−0.04	0.11 *	0.07	0.15 *	−0.05	−0.06	0.31 **	0.23 **	0.17 **	0.21 **	0.48 **	0.65 **	0.48 **	(0.86)	
Work withdrawal	11.16	4.55	0.18 **	−0.02	−0.18 *	−0.16 *	−0.03	−0.07	−0.50 **	−0.45 **	−0.43 **	−0.47 **	−0.21 **	−0.19 **	−0.21 **	−0.20 **	(0.84)

Note: ** *p* < 0.01; * *p* < 0.05. Cronbach’s alphas shown in the diagonal values that are in brackets.

**Table 2 ijerph-19-12575-t002:** Association of availability and utilization of family-friendly policies with work attitude and behavior through work–life conflict and perceived support of organization.

		Work–Life Conflict (WLC)				Perceived Supportive Organization (PSO)											
Antecedent		Coeff.	SE	*p*		Coeff.	SE	*p*									
FFP	a1	−0.007	0.1171	0.952	a2	0.26 *	0.1	0.01									
WLC					d21	−0.1 *	0.04	0.02									
PSO																	
Constant	iM1	3.61	0.07	<0.001	iM2	4.18	0.17	<0.001									
		Job Satisfaction				Affective Commitment			Intention to Stay			Job Engagement					
Antecedent		Coeff.	SE	*p*		Coeff.	SE	*p*	Coeff.	SE	*p*	Coeff.	SE	*p*			
FFP	c’	−0.06	0.09	0.5		0.12	0.08	0.13	0.09	0.08	0.24	0.27 *	0.1	0.007			
WLC	b1	−0.19	0.09	0.49		−0.16 *	0.03	<0.001	−0.19 *	0.03	<0.001	−0.08	0.04	0.054			
PSO	b2	0.11 *	0.04	0.01		0.2 *	0.03	<0.001	0.18 *	0.37	<0.001	0.34 *	0.05	<0.001			
Constant	iY	5.52	0.24	<0.001		4.2	0.21	<0.001	3.37	0.2	<0.001	3.11	0.25	<0.001			
		Task Performance				OCB(I)			OCB(O)			Citizenship Behavior			Work Withdrawal		
Antecedent		Coeff.	SE	*p*		Coeff.	SE	*p*	Coeff.	SE	*p*	Coeff.	SE	*p*	Coeff.	SE	*p*
FFP	C’	−0.13	0.08	0.11		−0.07	0.07	0.3	−0.06	0.07	0.43	−0.16 *	0.08	0.04	0.035	0.06	0.54
WLC	b1	−0.03	0.04	0.33		0.02	0.03	0.56	0.014	0.03	0.663	−0.01	0.03	0.69	0.07 *	0.02	0.004
PSO	b2	0.09 *	0.04	0.03		0.09 *	0.03	0.006	0.16 *	0.04	<0.001	0.06	0.04	0.69	−0.93 *	0.28	0.008
Constant	iY	5.45	0.21	<0.001		3.11	0.18	<0.001	2.6	0.19	<0.001	3.48	0.2	<0.001	2.5	0.15	<0.001

Note: * Significant at the *p* = 0.05 level.

**Table 3 ijerph-19-12575-t003:** Association of availability and utilization of family-friendly policies with work attitude and behavior through increased control of working hours.

		Control over Working Time (CWT)				Perceived Supportive Organization (PSO)											
Antecedent		Coeff.	SE	*p*		Coeff.	SE	*p*									
FFP	a1	0.42 *	0.06	<0.001	a2	0.08	0.11	0.48									
CWT					d21	0.44 *	0.09	<0.001									
PSO																	
Constant	iM1	2.6	0.35	<0.001	iM2	2.68	0.24	<0.001									
		Job Satisfaction				Affective Commitment			Intention to Stay			Job Engagement					
Antecedent		Coeff.	SE	*p*		Coeff.	SE	*p*	Coeff.	SE	*p*	Coeff.	SE	*p*			
FFP	c’	−0.17	0.1	0.09		−0.008	0.09	0.93	0	0.08	0.74	0.19	0.1	0.06			
CWT	b1	0.26 *	0.09	0.002		0.33 *	0.07	<0.001	0.29 *	0.07	<0.001	0.18 *	0.09	0.04			
PSO	b2	0.1 *	0.05	0.03		0.18 *	0.04	<0.001	0.17 *	0.04	<0.001	0.32 *	0.05	<0.001			
Constant	iY	4.16	0.25	<0.001		2.9	0.22	<0.001	1.98	0.21	<0.001	2.4	0.27	<0.001			
		Task Performance				OCB(I)			OCB(O)			Citizenship Behavior			Work Withdrawal		
Antecedent		Coeff.	SE	*p*		Coeff.	SE	*p*	Coeff.	SE	*p*	Coeff.	SE	*p*	Coeff.	SE	*p*
FFP	c’	−0.1	0.09	0.25		−0.1	0.07	0.17	−0.08	0.08	0.28	−0.23 *	0.08	0.005	0.03	0.06	0.64
CWT	b1	−0.08	0.08	0.26		0.07	0.06	0.24	0.07	0.07	0.29	0.19 *	0.07	0.007	0.02	0.053	0.71
PSO	b2	0.1 *	0.04	0.01		0.08 *	0.03	0.02	0.15 *	0.04	<0.001	0.04	0.04	0.35	−0.11 *	0.03	<0.001
Constant	iY	5.48	0.22	<0.001		3.03	0.18	<0.001	2.5	0.2	<0.001	3.02	0.21	<0.001	2.7	0.16	<0.001

Note: * Significant at the *p* = 0.05 level.

**Table 4 ijerph-19-12575-t004:** Interaction of FFP provision and use with supportive supervisors, and association with work–life conflict and control over working hours.

		Work–Life Conflict			Control over Working Time		
Antecedent		Coeff.	SE	*p*	Coeff.	SE	*p*
FFP	a1	−0.9	0.62	0.13	0.49	0.31	0.11
SS (Supportive Supervisors)	f1	−0.80 *	0.11	<0.001	0.19 *	0.05	0.005
FFP * SS	f2	0.28	0.18	0.12	−0.02	0.09	0.79
Constant	iM1	6.25	0.37	<0.001	1.94	0.19	<0.001

Note: * Significant at the *p* = 0.05 level.

## Data Availability

The data presented in this study are openly available in Public Sector Information (PSI) Portal at data.gov.hk, reference number 2014.A5.007.15A.

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
