# Peer review of "Family-Friendly Policies: Extrapolating A Pathway towards Better Work Attitudes and Work Behaviors in Hong Kong"

_ijerph, 2022, doi:10.3390/ijerph191912575_

Round 1

Reviewer 1 Report

Dear Authors,

Congratulations for your interesting and pertinent work.

In my opinion, and in order to be able to be published, small improvements are needed, namely:

- On page 8, at the end of the paragraph before table 1., please correct the error. 

Please review the writing of the text, as there are some spelling / English errors.

- Figures and tables should be placed in the text close to the first time they are referred to.

- In the conclusion section, authors should better clarify and highlight the main theoretical and practical contributions of their study this research field. 

Reviewer 2 Report

In the initial part of the document, before the method, the terms are even redundantly repeated in the wording: conflict between work and family, it is suggested to use synonyms, substitute wording, related terms, etc.

Review Writing Online 145, Studies-Studied

The findings are interesting, however, it is important to limit and perhaps analyze from the point of view of the type of company, because both the policies and the conflicts in the aforementioned relationship can be decisive in its characteristics.
